# Multiphysics Simulator for the IPMC Actuator: Mathematical Model, Finite Difference Scheme, Fast Numerical Algorithm, and Verification

**DOI:** 10.3390/mi11121119

**Published:** 2020-12-17

**Authors:** Anton P. Broyko, Ivan K. Khmelnitskiy, Eugeny A. Ryndin, Andrey V. Korlyakov, Nikolay I. Alekseyev, Vagarshak M. Aivazyan

**Affiliations:** Department of Micro- and Nanoelectronics, Faculty of Electronics, Saint Petersburg Electrotechnical University “LETI”, 5a Professor Popov St., Building 5, Saint Petersburg 197376, Russia; broyko@gmail.com (A.P.B.); khmelnitskiy@gmail.com (I.K.K.); akorl@yandex.ru (A.V.K.); NIAlekseyev@yandex.ru (N.I.A.); vaghik007@mail.ru (V.M.A.)

**Keywords:** IPMC actuator, cantilever beam, tip displacement, mechanical oscillations, finite difference scheme, algorithm for numerical simulation

## Abstract

The article is devoted to the development and creation of a multiphysics simulator that can, on the one hand, simulate the most significant physical processes in the IPMC actuator, and on the other hand, unlike commercial products such as COMSOL, can use computing resources economically. The developed mathematical model is an adjoint differential equation describing the transport of charged particles and water molecules in the ion-exchange membrane, the electrostatic field inside, and the mechanical deformation of the actuator. The distribution of the electrostatic potential in the interelectrode space is located by means of the solution of the Poisson equation with the Dirichlet boundary conditions, where the charge density is a function of the concentration of cations inside the membrane. The cation distribution was obtained by means of the solution of the equation system, in which the fluxes of ions and water molecules are described by the modified Nernst-Planck equations with boundary conditions of the third kind (the Robin problem). The cantilever beam forced oscillation equation in the presence of resistance (allowing for dissipative processes) with assumptions of elasticity theory was used to describe the actuator motion. A combination of the following computational methods was used as a numerical algorithm for the solution: the Poisson equation was solved by a direct method, the modified Nernst-Planck equations were solved by the Newton-Raphson method, and the mechanical oscillation equation was solved using an explicit scheme. For this model, a difference scheme has been created and an algorithm has been described, which can be implemented in any programming language and allows for fast computational experiments. On the basis of the created algorithm and with the help of the obtained experimental data, a program has been created and the verification of the difference scheme and the algorithm has been performed. Model parameters have been determined, and recommendations on the ranges of applicability of the algorithm and the program have been given.

## 1. Introduction

The development of propulsors for industrial and medical microrobots is an urgent task. Due to the requirements for such propulsors (workability, energy efficiency, the possibility of creating large forces and displacements in space), ionic polymer-metal composites (IPMCs) attract special attention. IPMCs are a sandwich structure consisting of a porous ion-exchange membrane impregnated with electrolyte and coated on both sides with metal electrodes [1,2].

The main inherent properties of IPMCs are flexibility, light weight, ease of manufacture, and processing. In addition, IPMCs are capable of large deformation in response to a voltage applied to the metallization of several volts. Due to this, they can be used as soft robotic actuators, artificial muscles, and dynamic sensors in the field of bionic engineering [3].

The operating principle of an IPMC actuator is based on transport processes in a polymer membrane. During water saturation, the dry polymer is structured in such a way that the hydrophilic ends of the polymer chains are oriented towards the water-filled membrane pores. Under the action of an electric field caused by the voltage applied to the electrodes, the charged liquid component in the membrane moves along the through pore system. The resulting electroosmotic water flow causes an increase in the liquid pressure at one electrode and a decrease in the vicinity of the other electrode. The pressure difference leads to bending of the IPMC actuator [1,4].

There are numerous attempts to describe the actuation mechanism of IPMCs. Since the middle of the last century, many researchers have been trying to simulate actuators and sensors based on IPMC. To study the behavior and predict the motion of an IPMC actuator, it is necessary to have reliable models. Currently, in order to describe the physical processes in the IPMC actuator, a large number of models have been developed, which are conventionally divided into three groups: multiphysics models (white box models), which describe the behavior of the problem parameter fields—such as ion concentration, potential, and deformation [5,6]—models based on the method of analogies or models with lumped parameters (grey box models) [7,8], and empirical models (black box models), which are approximation curves of experimental data [9]. In [6], a continuum electromechanical model is proposed to improve the production process and the composition of the IPMC. In [10], the authors presented the IPMC electromechanical reaction model, based on the electrostatic forces of attraction/repulsion in the IPMC, to predict the behavior of actuators, in particular, to explain the relaxation of the IPMC. These models can predict the bending behavior of the IPMC with a sufficient degree of accuracy, but they require many predetermined physicochemical properties that are measured in experiments. Black box models, also called empirical and phenomenological models—presented in [11]—are applicable only to specific shapes and boundary conditions. A more reasonable black box model was introduced in [12]. The new model was validated using the experimental results from some materials [13]. Grey box models proposed at [7,8] are based on physical laws and parameters determined by experiments.

Despite the long research time and a large number of different approaches, none of the models has proven its applicability to IPMCs with different thicknesses and different compositions of the ionic polymer membrane [14].

Further development led to the creation of a model that explicitly took into account the solvent (water) dynamics [15]. This model has a significant advantage of being able to describe the dynamic response that is highly dependent on the IPMC moisture saturation. However, it is difficult to perform analysis and simulation [16], since the mathematical model presented in [15] consists of complex nonlinear partial differential equations. We have made an attempt to solve this problem by creating a calculation scheme and a program code. A combination of the following computational methods was used as a numerical algorithm for the solution: the Poisson equation was solved by a direct method, the modified Nernst-Planck equations were solved by the Newton-Raphson method, and the mechanical oscillation equation was solved using an explicit scheme.

## 2. Mathematical Model

Taking into account the complex nature of non-stationary processes in the considered IPMC actuator, the developed mathematical model of the actuator is based on the thermodynamic theory of irreversible processes and includes the modified Nernst-Planck equations [15], the Poisson equation [17] and the mechanical oscillation equation of a cantilever beam with a fixed end [18,19].

The modified Nernst-Planck equations considered in [15] describe the transfer processes of ions and water molecules in the IPMC membrane with the following assumptions:

Ionic polymer-metal composite is considered to be two-phase and includes the solid phase, which is a polymer porous structure, fixed negative charge and metal electrodes, and the liquid phase, which includes cations and water molecules, redistributed under an electric field and/or a mechanical load.The liquid phase flux consists of two components: diffusion (including electromigration) and convective. The diffusion fluxes of ions and water molecules are determined by the potential gradient, the concentration gradients of ions and water molecules, and the hydrostatic pressure gradient created by redistribution of ions and water molecules in polymer nanopores. The solid phase influences the diffusion fluxes through the nanopore structure and the electric field of fixed negative ions in the membrane. The convective fluxes are determined by the elastic force of the solid phase.In a short time interval, the hydraulic pressure and the inherent mechanical stress are balanced with the elastic stress of the composite solid phase.

As applied to the problem under consideration, the modified Nernst-Planck equations can be written as [15]
(1)$$J_{I}=-D_{II}\left(\frac{\partial C_{I}}{\partial y}+\frac{C_{I}}{RT}\left(Z_{I}F\frac{\partial \varphi }{\partial y}+V_{I}\frac{\partial p}{\partial y}\right)\right)-D_{II}n_{dW}\frac{C_{I}}{C_{W}}\left(\frac{\partial C_{W}}{\partial y}+\frac{C_{W}V_{W}}{RT}\cdot \frac{\partial p}{\partial y}\right)-C_{I}K\frac{\partial p}{\partial y};$$
(2)$$J_{W}=-D_{WW}\left(\frac{\partial C_{W}}{\partial y}+\frac{C_{W}V_{W}}{RT}\cdot \frac{\partial p}{\partial y}\right)-D_{II}n_{dW}\left(\frac{\partial C_{I}}{\partial y}+\frac{C_{I}}{RT}\left(Z_{I}F\frac{\partial \varphi }{\partial y}+V_{I}\frac{\partial p}{\partial y}\right)\right)-C_{W}K\frac{\partial p}{\partial y};$$
where $J_{I}$ is the flux density of cations in the polymer volume; $J_{W}$ is the flux density of water molecules in the polymer volume; $C_{I}$ is the concentration of cations in the polymer volume; $C_{W}$ is the concentration of water molecules in the polymer volume; p is the hydraulic pressure in the polymer; $V_{I}$ is the molar volume of ions; $V_{W}$ is the molar volume of water; $n_{dW}$ is the number of water molecules associated with one cation; $D_{II}$ is the self-diffusion coefficient of cations; $D_{WW}$ is the self-diffusion coefficient of water; K is the filtration coefficient (Darcy’s law); φ is the electrostatic potential; $Z_{I}$ is the relative charge of ion; T is the absolute temperature; F is the Faraday constant; R is the gas constant; y is the coordinate.

The summands $-D_{II}\left(\frac{\partial C_{I}}{\partial y}+\frac{C_{I}}{RT}\left(Z_{I}F\frac{\partial \varphi }{\partial y}+V_{I}\frac{\partial p}{\partial y}\right)\right)$ and $-D_{WW}\left(\frac{\partial C_{W}}{\partial y}+\frac{C_{W}V_{W}}{RT}\cdot \frac{\partial p}{\partial y}\right)$ in Equations (1), (2) represent the self-diffusion components of the flux density, initiated by the potential gradient, the concentration gradients of cations and water molecules, and the hydrostatic pressure gradient.

The summands $-D_{II}n_{dW}\left(\frac{\partial C_{I}}{\partial y}+\frac{C_{I}}{RT}\left(Z_{I}F\frac{\partial \varphi }{\partial y}+V_{I}\frac{\partial p}{\partial y}\right)\right)$ and $-D_{II}n_{dW}\frac{C_{I}}{C_{W}}\left(\frac{\partial C_{W}}{\partial y}+\frac{C_{W}V_{W}}{RT}\cdot \frac{\partial p}{\partial y}\right)$ reflect the interaction of the diffusion fluxes of cations and water molecules in the IPMC. Since a cation carries $n_{dW}$ water molecules by hydration, the diffusion coefficients of the interaction between the fluxes of cations and water $D_{WI}$ and, accordingly, $D_{IW}$ are expressed in these summands through the self-diffusion coefficient of ions as
(3)$$D_{WI}=D_{II}n_{dW};$$
(4)$$D_{IW}=D_{II}n_{dW}\frac{C_{I}}{C_{W}}.$$

The summands $-C_{I}K\frac{\partial p}{\partial y}$ and $-C_{W}K\frac{\partial p}{\partial y}$ describe, using Darcy’s law [15], the convective components of the flux density, which determine the relationship between the hydraulic pressure of the liquid phase and the elastic mechanical stress of the solid phase of IPMC.

In order to study the dynamics of changes in the spatial distributions of the concentrations of cations and water molecules in time, taking into account that the rates of changes in the concentrations of ions and water molecules in time are determined by the divergence of the corresponding flux densities
(5)$$\frac{\partial C_{I}}{\partial t}=\frac{\partial J_{I}}{\partial y};$$
(6)$$\frac{\partial C_{W}}{\partial t}=\frac{\partial J_{W}}{\partial y},$$
where t is time, the Nernst-Planck Equations (1) and (2) can be written as
(7)$$\frac{\partial C_{I}}{\partial t}=\frac{\partial }{\partial y}\left[D_{II}\left(\frac{\partial C_{I}}{\partial y}+\frac{C_{I}}{RT}\left(Z_{I}F\frac{\partial \varphi }{\partial y}+V_{I}\frac{\partial p}{\partial y}\right)\right)+D_{II}n_{dW}\frac{C_{I}}{C_{W}}\left(\frac{\partial C_{W}}{\partial y}+\frac{C_{W}V_{W}}{RT}\cdot \frac{\partial p}{\partial y}\right)+C_{I}K\frac{\partial p}{\partial y}\right];$$
(8)$$\frac{\partial C_{W}}{\partial t}=\frac{\partial }{\partial y}\left[D_{WW}\left(\frac{\partial C_{W}}{\partial y}+\frac{C_{W}V_{W}}{RT}\cdot \frac{\partial p}{\partial y}\right)+D_{II}n_{dW}\left(\frac{\partial C_{I}}{\partial y}+\frac{C_{I}}{RT}\left(Z_{I}F\frac{\partial \varphi }{\partial y}+V_{I}\frac{\partial p}{\partial y}\right)\right)+C_{W}K\frac{\partial p}{\partial y}\right].$$

The concentration of cations at the initial time $C_{I}\left(y,t_{\min}\right)$ (the initial condition for Equation (7)) is determined by the initial concentration of ions in the polymer $C_{+}$ in accordance with the expression
(9)$$C_{I}\left(y,t_{\min}\right)=C_{+}\frac{\rho_{SPN}\left(1-P_{WN}\right)}{{\left(1+\alpha \right)}^{3}h},$$
where $\rho_{SPN}$ is the layer density of the dry membrane at normal ambient humidity; h is the thickness of the dry membrane (excluding metal electrodes) at normal ambient humidity; α is the expansion coefficient of the membrane at maximum humidification; $P_{WN}$ is the mass fraction of water in the dry polymer at normal ambient humidity.

The boundary conditions for Equation (7) determine the ion flux densities through the outer boundaries of the polymer, caused by the evaporation processes
(10)$$\begin{array}{l} {\left.\left\{D_{II}\left(\frac{\partial C_{I}}{\partial y}+\frac{C_{I}}{RT}\left(Z_{I}F\frac{\partial \varphi }{\partial y}+V_{I}\frac{\partial p}{\partial y}\right)\right)+D_{II}n_{dW}\frac{C_{I}}{C_{W}}\left(\frac{\partial C_{W}}{\partial y}+\frac{C_{W}V_{W}}{RT}\cdot \frac{\partial p}{\partial y}\right)+C_{I}K\frac{\partial p}{\partial y}\right\}\right|}_{\left(y_{\min},t\right)}=\\ =\gamma_{I}\frac{D_{II}}{H}C_{I}\left(y_{\min},t\right); \end{array}$$
(11)$$\begin{array}{l} {\left.\left\{D_{II}\left(\frac{\partial C_{I}}{\partial y}+\frac{C_{I}}{RT}\left(Z_{I}F\frac{\partial \varphi }{\partial y}+V_{I}\frac{\partial p}{\partial y}\right)\right)+D_{II}n_{dW}\frac{C_{I}}{C_{W}}\left(\frac{\partial C_{W}}{\partial y}+\frac{C_{W}V_{W}}{RT}\cdot \frac{\partial p}{\partial y}\right)+C_{I}K\frac{\partial p}{\partial y}\right\}\right|}_{\left(y_{\max},t\right)}=\\ =-\gamma_{I}\frac{D_{II}}{H}C_{I}\left(y_{\max},t\right), \end{array}$$
where $y_{\min},y_{\max}$ are the coordinates of the polymer membrane boundaries; H is the thickness of metal electrodes; $\gamma_{I}$ is the dimensionless empirical coefficient determining the evaporation rate of cations into the external environment.

The concentration of water molecules at the initial time $C_{W}\left(y,t_{\min}\right)$ (the initial condition for Equation (8)) is determined by the degree of membrane humidification $K_{w}$ (the ratio of the concentration of water molecules in the polymer to the maximum possible concentration) in accordance with the expression
(12)$$C_{W}\left(y,t_{\min}\right)=K_{w}\frac{\rho_{SPN}\left(1-P_{WN}\right)P_{WS}}{M_{W}{\left(1+\alpha \right)}^{3}h},$$
where $M_{W}$ is the molar mass of water; $P_{WS}$ is the mass fraction of water in the maximum humidified polymer.

The boundary conditions for Equation (8) determine the fluxes of water molecules evaporated through the outer boundaries of the polymer
(13)$$\begin{array}{l} {\left.\left\{D_{WW}\left(\frac{\partial C_{W}}{\partial y}+\frac{C_{W}V_{W}}{RT}\cdot \frac{\partial p}{\partial y}\right)+D_{II}n_{dW}\left(\frac{\partial C_{I}}{\partial y}+\frac{C_{I}}{RT}\left(Z_{I}F\frac{\partial \varphi }{\partial y}+V_{I}\frac{\partial p}{\partial y}\right)\right)+C_{W}K\frac{\partial p}{\partial y}\right\}\right|}_{\left(y_{\min},t\right)}=\\ =\gamma_{W}\frac{D_{WW}}{H}C_{W}\left(y_{\min},t\right); \end{array}$$
(14)$$\begin{array}{l} {\left.\left\{D_{WW}\left(\frac{\partial C_{W}}{\partial y}+\frac{C_{W}V_{W}}{RT}\cdot \frac{\partial p}{\partial y}\right)+D_{II}n_{dW}\left(\frac{\partial C_{I}}{\partial y}+\frac{C_{I}}{RT}\left(Z_{I}F\frac{\partial \varphi }{\partial y}+V_{I}\frac{\partial p}{\partial y}\right)\right)+C_{W}K\frac{\partial p}{\partial y}\right\}\right|}_{\left(y_{\max},t\right)}=\\ =-\gamma_{W}\frac{D_{WW}}{H}C_{W}\left(y_{\max},t\right), \end{array}$$
where $\gamma_{W}$ the dimensionless empirical coefficient determining the evaporation rate of water molecules into the external environment.

The introduction of the empirical coefficients $\gamma_{I},\;\gamma_{W}$ into boundary conditions (10), (11), (13), (14) is due to the complexity of theoretical evaluation of the evaporation rates of water molecules and ions. This is due, in particular, to the considerable variation in the parameters of the granular structure of metal electrodes.

The Poisson equation connects the spatial distributions of the potential and the electric field in the IPMC membrane with the voltage at the electrodes and the spatial distribution of ions [17]
(15)$$\frac{\partial^{2}\varphi }{\partial y^{2}}=-\frac{qZ_{I}F}{\varepsilon \varepsilon_{0}}\left(C_{I}-C_{-}\right),$$
where $C_{-}$ is the concentration of anions; q is the elementary charge; ε is the relative permittivity of water; $\varepsilon_{0}$ is the permittivity of vacuum.

A uniform stationary spatial distribution of the concentration of anions is assumed
(16)$$C_{-}=C_{+}\frac{\rho_{SPN}\left(1-P_{WN}\right)}{{\left(1+\alpha \right)}^{3}h}.$$

The boundary conditions for the Poisson Equation (15) are determined by the potentials at the electrodes
(17)$$\varphi \left(y_{\min},t\right)=0;$$
(18)$$\varphi \left(y_{\max},t\right)=U\left(t\right),$$
where $U\left(t\right)$ is the time dependence of the voltage applied to the electrodes.

The mechanical oscillation equation of a cantilever beam with a distributed mass, taking into account attenuation, describes the tip displacement of the beam relative to the equilibrium position under the action of an exciting force created by the transport of ions and water molecules and their interaction with the composite solid phase [18,19]
(19)$$\frac{\partial^{2}s\left(t\right)}{\partial t^{2}}+\beta \frac{\partial s\left(t\right)}{\partial t}+\omega_{0}^{2}s\left(t\right)=\frac{F_{L}\left(t\right)}{m_{L}},$$
where $s\left(t\right)$ is the tip displacement of the beam relative to the equilibrium position; $m_{L}$ is the linear density of the cantilever beam; $F_{L}\left(t\right)$ is the exciting force per unit of beam length; $\omega_{0}$ is the natural oscillation frequency of the cantilever beam; β is the coefficient characterizing dissipative processes.

For the cantilever beam stationary bending, corresponding to the tip displacement $s_{0}$ of the beam relative to the equilibrium position, the right-hand side of Equation (19) can be expressed through the natural oscillation frequency $\omega_{0}$ of the cantilever beam as follows
(20)$$\frac{F_{L}}{m_{L}}=\frac{k}{m_{L}L_{S}}s_{0}=\omega_{0}^{2}s_{0},$$
where $L_{S}=L\left(1+\alpha \right)$ is the length of the humidified beam; k is the stiffness coefficient of the cantilever beam.

Taking into account that the bending moment is created by the transport of ions and water molecules and their interaction with the composite solid phase is approximately uniform along the beam length, the tip displacement of the beam relative to the equilibrium position can be expressed through the bending moment as [19]
(21)$$s_{0}=\frac{L_{S}{}^{2}}{2E_{eq}J_{y}}M,$$
where M is the bending moment; $E_{eq}$ is the equivalent Young’s modulus of a three-layer cantilever beam (polymer membrane with electrodes); $J_{y}$ is the moment of inertia of the beam section.

Then, substituting (20) and (21) into the right-hand side of Equation (19), we obtain
(22)$$\frac{\partial^{2}s\left(t\right)}{\partial t^{2}}+\beta \frac{\partial s\left(t\right)}{\partial t}+\omega_{0}^{2}s\left(t\right)=\frac{\omega_{0}^{2}L_{S}{}^{2}}{2E_{eq}J_{y}}M\left(t\right).$$

The bending moment $M\left(t\right)$ is determined by the spatial distribution of the hydraulic pressure $p\left(y,t\right)$ in accordance with the expression
(23)$$M\left(t\right)=w_{S}\overset{h_{S}/2}{\underset{-h_{S}/2}{\int }}\left(p\left(y,t\right)-\frac{1}{h}\overset{h_{S}/2}{\underset{-h_{S}/2}{\int }}p\left(y,t\right)dy\right)ydy,$$
where $w_{S}=w\left(1+\alpha \right)$ is the width of the humidified beam; $h_{S}=h\left(1+\alpha \right)$ is the thickness of the humidified polymer. The integral $\frac{1}{h}\overset{h_{S}/2}{\underset{-h_{S}/2}{\int }}p\left(y,t\right)dy$ in expression (23) determines the average value of the hydraulic pressure in the beam at the current time.

The spatial distribution of the hydraulic pressure at the current time can be expressed through the spatial distributions of the concentrations of water molecules and ions using the linear relation [1,20] as
(24)$$p\left(y,t\right)=\eta_{I}V_{I}\left(C_{I}\left(y,t\right)-C_{I}\left(t_{\min}\right)\right)+\eta_{W}V_{W}\left(C_{W}\left(y,t\right)-C_{W}\left(t_{\min}\right)\right),$$
where $t_{\min}$ is the initial time; $\eta_{I},\;\eta_{W}$ is the empirical coefficients having pressure units.

The product of the equivalent Young’s modulus of the cantilever beam and the moment of inertia of the section for the problem under consideration is expressed by the integral
(25)$$E_{eq}J_{y}=w_{S}\overset{\frac{h_{S}}{2}+H}{\underset{-\frac{h_{S}}{2}-H}{\int }}E\left(y,t\right)y^{2}dy.$$

In expression (25), the dependence of the Young’s modulus on the coordinate $E\left(y,t\right)$ at the current time is considered not only in the polymer, but also in metal electrodes, as evidenced by the integration limits $\pm \left(\frac{h_{S}}{2}+H\right).$ In this case, the dependence of the Young’s modulus on the concentration of water molecules in the polymer is taken into account, which is determined experimentally in accordance with the procedure described in [21].

After substituting (23), (24), and (25) into (22), we obtain the cantilever beam mechanical oscillation equation in the form
(26)$$\begin{array}{c} \frac{\partial^{2}s\left(t\right)}{\partial t^{2}}+\beta \frac{\partial s\left(t\right)}{\partial t}+\omega_{0}^{2}s\left(t\right)=\\ =\frac{\omega_{0}^{2}L_{S}{}^{2}}{2\int_{-\frac{h_{S}}{2}-H}^{\frac{h_{S}}{2}+H}E\left(C_{W}\left(y,t\right)\right)y^{2}dy}\left[\eta_{I}V_{I}\overset{\frac{h_{S}}{2}}{\underset{-\frac{h_{S}}{2}}{\int }}\left(C_{I}\left(y,t\right)-\frac{1}{h_{S}}\overset{\frac{h_{S}}{2}}{\underset{-\frac{h_{S}}{2}}{\int }}C_{I}\left(y,t\right)dy\right)ydy+\right.\\ \left.+\eta_{W}V_{W}\overset{h_{S}/2}{\underset{-h_{S}/2}{\int }}\left(C_{W}\left(y,t\right)-\frac{1}{h_{S}}\overset{h_{S}/2}{\underset{-h_{S}/2}{\int }}C_{W}\left(y,t\right)dy\right)ydy\right]. \end{array}$$

The initial conditions for Equation (26) determine zero values of the tip displacement and the tip velocity of the beam at the initial time
(27)$$s\left(t_{\min}\right)=0;$$
(28)$${\left.\frac{\partial s}{\partial t}\right|}_{t_{\min}}=0.$$

The resonant frequency $\omega_{0}$ of mechanical oscillations of the beam with a distributed mass in Equation (26) is determined as [19]
(29)$$\omega_{0}\left(t\right)=\frac{\lambda }{L_{S}{}^{2}}\sqrt{\frac{E_{eq}J_{y}}{m_{L}\left(t\right)}},$$
where λ is the coefficient depending on the mode of beam bending oscillations.

The linear density of the beam $m_{L}\left(t\right)$ is determined as the sum of the linear densities of the polymer, electrodes and water in the polymer and, taking into account the water evaporation process, is a function of time
(30)$$m_{L}\left(t\right)=\frac{m_{D}}{L_{S}}+2\rho_{M}w_{S}H+M_{W}w_{S}\overset{\frac{h_{S}}{2}+H}{\underset{-\frac{h_{S}}{2}-H}{\int }}C_{W}\left(y,t\right)dy,$$
where $m_{D}$ is the mass of the beam dry polymer; $\rho_{M}$ is the density of the electrode material.

Substituting expressions (25) and (30) into (29), we obtain
(31)$$\omega_{0}\left(t\right)=\frac{\lambda }{L_{S}{}^{2}}\sqrt{\frac{w_{S}\int_{-\frac{h_{S}}{2}-H}^{\frac{h_{S}}{2}+H}E\left(y,t\right)y^{2}dy}{\frac{m_{D}}{L_{S}}+2\rho_{M}w_{S}H+M_{W}w_{S}\int_{-\frac{h_{S}}{2}-H}^{\frac{h_{S}}{2}+H}C_{W}\left(y,t\right)dy}}.$$

In this study, the system of Equations (7), (8), (15), and (26) with initial conditions (9), (12), (27), (28); boundary conditions (10), (11), (13), (14), (17) and (18); and parameters (16) and (31) was solved numerically using the finite difference method without additional simplifications.

## 3. Discretization of the Mathematical Model, Numerical Simulation Technique

The discretization of the model described above was carried out within the framework of the finite difference method on uniform time $G_{t}$ and coordinate $G_{y}$ grids
(32)$$G_{t}=\left\{t_{m}=\left(m-1\right)\Delta t\left|m=1,\ldots ,M\right.\right\};$$
(33)$$G_{y}=\left\{y_{j}=\left(j-1\right)\Delta y\left|j=1,\ldots ,J\right.\right\},$$
where Δt is the grid step in time; Δy is the coordinate grid step; m is the point index $t_{m}$ of the time grid; j is the point index $y_{j}$ of the coordinate grid; M is the number of points in the time grid; J is the number of points in the coordinate grid. Moreover, the coordinate grid (33) covers only the polymer part of the three-layer beam between the boundaries with metal electrodes.

In order to optimize the simulation time on a personal computer with insignificant RAM resources (8 GB), a self-consistent numerical solution of three subsystems of the proposed model at each time slice was carried out sequentially using a combination of the following methods:Subsystem 1, which includes Poisson Equation (15) with boundary conditions (17), (18), was solved by a direct methodSubsystem 2, which includes modified Nernst-Planck Equations (7) and (8) with initial conditions (9) and (12) and boundary conditions (10), (11), (13), and (14), was solved using the Newton-Raphson methodSubsystem 3, which includes cantilever beam mechanical oscillation Equation (26) with initial conditions (27) and (28), was solved using an explicit scheme

Taking into account the listed methods for the numerical solution of the proposed model equations, discretization schemes for subsystems 1–3 on time and coordinate grids (32) and (33) are obtained in the following form:
Subsystem 1 (15), (17), (18)
(34)$$\frac{\varphi_{j+1}^{m}-2\varphi_{j}^{m}+\varphi_{j-1}^{m}}{\Delta y^{2}}=-\frac{qZ_{I}F}{\varepsilon \varepsilon_{0}}\left(C_{I}{}_{j}^{m}-C_{-}\right);\;$$
(35)$$\varphi_{1}^{m}=0;$$
(36)$$\varphi_{J}^{m}=U_{m},$$
where $\varphi_{j}^{m}$ is the grid function of the potential; $C_{I}{}_{j}^{m}$ is the grid function of the concentration of cations; $U_{m}$ is the voltage applied to the electrodes at the time point $t_{m}$.Subsystem 2 (7)–(14)
(37)$$\begin{array}{ll} \; & {C_{I}}_{j}^{m+1}-\frac{D_{II}\Delta t}{\Delta y^{2}}\left[{C_{I}}_{j+1}^{m+1}-2{C_{I}}_{j}^{m+1}+{C_{I}}_{j-1}^{m+1}\right.+\\ \; & +\frac{Z_{I}F}{2RT}\left(\left({C_{I}}_{j+1}^{m+1}+{C_{I}}_{j}^{m+1}\right)\left(\varphi_{j+1}^{m+1}-\varphi_{j}^{m+1}\right)-\left({C_{I}}_{j}^{m+1}+{C_{I}}_{j-1}^{m+1}\right)\left(\varphi_{j}^{m+1}-\varphi_{j-1}^{m+1}\right)\right)+\\ \; & +\frac{n_{dW}}{2}\left(\left({C_{I}}_{j+1}^{m+1}+{C_{I}}_{j}^{m+1}\right)\left(\ln{C_{W}}_{j+1}^{m+1}-\ln{C_{W}}_{j}^{m+1}\right)-\left({C_{I}}_{j}^{m+1}+{C_{I}}_{j-1}^{m+1}\right)\left(\ln{C_{W}}_{j}^{m+1}-\ln{C_{W}}_{j-1}^{m+1}\right)\right)+\\ \; & +\frac{\eta_{I}V_{I}}{2}\left(\frac{V_{I}+n_{dW}V_{W}}{RT}+\frac{K}{D_{II}}\right)\left(\left({C_{I}}_{j+1}^{m+1}+{C_{I}}_{j}^{m+1}\right)\left({C_{I}}_{j+1}^{m+1}-{C_{I}}_{j}^{m+1}\right)\right.-\\ \; & -\left.\left({C_{I}}_{j}^{m+1}+{C_{I}}_{j-1}^{m+1}\right)\left({C_{I}}_{j}^{m+1}-{C_{I}}_{j-1}^{m+1}\right)\right)+\\ \; & +\frac{\eta_{W}V_{W}}{2}\left(\frac{V_{I}+n_{dW}V_{W}}{RT}+\frac{K}{D_{II}}\right)\left(\left({C_{I}}_{j+1}^{m+1}+{C_{I}}_{j}^{m+1}\right)\left({C_{W}}_{j+1}^{m+1}-{C_{W}}_{j}^{m+1}\right)\right.-\\ \; & -\left.\left.\left({C_{I}}_{j}^{m+1}+{C_{I}}_{j-1}^{m+1}\right)\left({C_{W}}_{j}^{m+1}-{C_{W}}_{j-1}^{m+1}\right)\right)\right]-{C_{I}}_{j}^{m}=0; \end{array}$$
(38)$$\begin{array}{ll} \; & {C_{W}}_{j}^{m+1}-\frac{\Delta t}{\Delta y^{2}}\left[D_{WW}\left({C_{W}}_{j+1}^{m+1}-2{C_{W}}_{j}^{m+1}+{C_{W}}_{j-1}^{m+1}\right)\right.+\\ \; & +\frac{\eta_{W}V_{W}}{2}\left(D_{WW}\frac{V_{W}}{RT}+K\right)\left(\left({C_{W}}_{j+1}^{m+1}+{C_{W}}_{j}^{m+1}\right)\left({C_{W}}_{j+1}^{m+1}-{C_{W}}_{j}^{m+1}\right)\right.-\\ \; & -\left.\left({C_{W}}_{j}^{m+1}+{C_{W}}_{j-1}^{m+1}\right)\left({C_{W}}_{j}^{m+1}-{C_{W}}_{j-1}^{m+1}\right)\right)+n_{dW}D_{II}\left({C_{I}}_{j+1}^{m+1}-2{C_{I}}_{j}^{m+1}+{C_{I}}_{j-1}^{m+1}\right)+\\ \; & +\eta_{I}n_{dW}\frac{D_{II}}{2}\frac{{V_{I}}^{2}}{RT}\left(\left({C_{I}}_{j+1}^{m+1}+{C_{I}}_{j}^{m+1}\right)\left({C_{I}}_{j+1}^{m+1}-{C_{I}}_{j}^{m+1}\right)-\left({C_{I}}_{j}^{m+1}+{C_{I}}_{j-1}^{m+1}\right)\left({C_{I}}_{j}^{m+1}-{C_{I}}_{j-1}^{m+1}\right)\right)+\\ \; & +\eta_{W}n_{dW}\frac{D_{II}}{2}\frac{V_{I}V_{W}}{RT}\left(\left({C_{I}}_{j+1}^{m+1}+{C_{I}}_{j}^{m+1}\right)\left({C_{W}}_{j+1}^{m+1}-{C_{W}}_{j}^{m+1}\right)-\left({C_{I}}_{j}^{m+1}+{C_{I}}_{j-1}^{m+1}\right)\left({C_{W}}_{j}^{m+1}-{C_{W}}_{j-1}^{m+1}\right)\right)+\\ \; & +\frac{\eta_{I}V_{I}}{2}\left(D_{WW}\frac{V_{W}}{RT}+K\right)\left(\left({C_{W}}_{j+1}^{m+1}+{C_{W}}_{j}^{m+1}\right)\left({C_{I}}_{j+1}^{m+1}-{C_{I}}_{j}^{m+1}\right)-\left({C_{W}}_{j}^{m+1}+{C_{W}}_{j-1}^{m+1}\right)\left({C_{I}}_{j}^{m+1}-{C_{I}}_{j-1}^{m+1}\right)\right)+\\ \; & +\left.n_{dW}\frac{D_{II}}{2}\frac{Z_{I}F}{RT}\left(\left({C_{I}}_{j+1}^{m+1}+{C_{I}}_{j}^{m+1}\right)\left(\varphi_{j+1}^{m+1}-\varphi_{j}^{m+1}\right)-\left({C_{I}}_{j}^{m+1}+{C_{I}}_{j-1}^{m+1}\right)\left(\varphi_{j}^{m+1}-\varphi_{j-1}^{m+1}\right)\right)\right]-{C_{W}}_{j}^{m}=0; \end{array}$$
(39)$$C_{I}{}_{j}^{1}=C_{+}\frac{\rho_{SPN}\left(1-P_{WN}\right)}{{\left(1+\alpha \right)}^{3}h};$$
(40)$$C_{W}{}_{j}^{1}=K_{w}\frac{\rho_{SPN}\left(1-P_{WN}\right)P_{WS}}{M_{W}{\left(1+\alpha \right)}^{3}h};$$
(41)$$\begin{array}{l} C_{I}{}_{2}^{m+1}-C_{I}{}_{1}^{m+1}+\frac{Z_{I}F}{RT}C_{I}{}_{1}^{m+1}\left(\varphi_{2}^{m+1}-\varphi_{1}^{m+1}\right)+n_{dW}C_{I}{}_{1}^{m+1}\left(\ln C_{W}{}_{2}^{m+1}-\ln C_{W}{}_{1}^{m+1}\right)+\\ +\eta_{I}V_{I}\left(\frac{V_{I}+n_{dW}V_{W}}{RT}+\frac{K}{D_{II}}\right)C_{I}{}_{1}^{m+1}\left(C_{I}{}_{2}^{m+1}-C_{I}{}_{1}^{m+1}\right)+\\ +\eta_{W}V_{W}\left(\frac{V_{I}+n_{dW}V_{W}}{RT}+\frac{K}{D_{II}}\right)C_{I}{}_{1}^{m+1}\left(C_{W}{}_{2}^{m+1}-C_{W}{}_{1}^{m+1}\right)-\Delta y\frac{\gamma_{I}}{H}C_{I}{}_{1}^{m+1}=0; \end{array}$$
(42)$$\begin{array}{l} C_{I}{}_{J}^{m+1}-C_{I}{}_{J-1}^{m+1}+\frac{Z_{I}F}{RT}C_{I}{}_{J}^{m+1}\left(\varphi_{J}^{m+1}-\varphi_{J-1}^{m+1}\right)+n_{dW}C_{I}{}_{J}^{m+1}\left(\ln C_{W}{}_{J}^{m+1}-\ln C_{W}{}_{J-1}^{m+1}\right)+\\ +\eta_{I}V_{I}\left(\frac{V_{I}+n_{dW}V_{W}}{RT}+\frac{K}{D_{II}}\right)C_{I}{}_{J}^{m+1}\left(C_{I}{}_{J}^{m+1}-C_{I}{}_{J-1}^{m+1}\right)+\\ +\eta_{W}V_{W}\left(\frac{V_{I}+n_{dW}V_{W}}{RT}+\frac{K}{D_{II}}\right)C_{I}{}_{J}^{m+1}\left(C_{W}{}_{J}^{m+1}-C_{W}{}_{J-1}^{m+1}\right)+\Delta y\frac{\gamma_{I}}{H}C_{I}{}_{J}^{m+1}=0; \end{array}$$(43)$$\begin{array}{cc} \; & D_{WW}\left({C_{W}}_{2}^{m+1}-{C_{W}}_{1}^{m+1}\right)+\eta_{W}V_{W}\left(D_{WW}\frac{V_{W}}{RT}+K\right){C_{W}}_{1}^{m+1}\left({C_{W}}_{2}^{m+1}-{C_{W}}_{1}^{m+1}\right)+\\ \; & +\eta_{I}V_{I}\left(D_{WW}\frac{V_{W}}{RT}+K\right){C_{W}}_{1}^{m+1}\left({C_{I}}_{2}^{m+1}-{C_{I}}_{1}^{m+1}\right)+D_{II}n_{dW}\left({C_{I}}_{2}^{m+1}-{C_{I}}_{1}^{m+1}\right)+\\ \; & +\eta_{W}n_{dW}D_{II}\frac{V_{I}V_{W}}{RT}{C_{I}}_{1}^{m+1}\left({C_{W}}_{2}^{m+1}-{C_{W}}_{1}^{m+1}\right)+\eta_{I}n_{dW}D_{II}\frac{{V_{I}}^{2}}{RT}{C_{I}}_{1}^{m+1}\left({C_{I}}_{2}^{m+1}-{C_{I}}_{1}^{m+1}\right)+\\ \; & +n_{dW}D_{II}\frac{Z_{I}F}{RT}{C_{I}}_{1}^{m+1}\left(\varphi_{2}^{m+1}-\varphi_{1}^{m+1}\right)-\Delta y\gamma_{W}\frac{D_{WW}}{H}{C_{W}}_{1}^{m+1}=0; \end{array}$$
(44)$$\begin{array}{cc} \; & D_{WW}\left({C_{W}}_{J}^{m+1}-{C_{W}}_{J-1}^{m+1}\right)+\eta_{W}V_{W}\left(D_{WW}\frac{V_{W}}{RT}+K\right){C_{W}}_{J}^{m+1}\left({C_{W}}_{J}^{m+1}-{C_{W}}_{J-1}^{m+1}\right)+\\ \; & +\eta_{I}V_{I}\left(D_{WW}\frac{V_{W}}{RT}+K\right){C_{W}}_{J}^{m+1}\left({C_{I}}_{J}^{m+1}-{C_{I}}_{J-1}^{m+1}\right)+D_{II}n_{dW}\left({C_{I}}_{J}^{m+1}-{C_{I}}_{J-1}^{m+1}\right)+\\ \; & +\eta_{W}n_{dW}D_{II}\frac{V_{I}V_{W}}{RT}{C_{I}}_{J}^{m+1}\left({C_{W}}_{J}^{m+1}-{C_{W}}_{J-1}^{m+1}\right)+\eta_{I}n_{dW}D_{II}\frac{{V_{I}}^{2}}{RT}{C_{I}}_{J}^{m+1}\left({C_{I}}_{J}^{m+1}-{C_{I}}_{J-1}^{m+1}\right)+\\ \; & +n_{dW}D_{II}\frac{Z_{I}F}{RT}{C_{I}}_{J}^{m+1}\left(\varphi_{J}^{m+1}-\varphi_{J-1}^{m+1}\right)+\Delta y\gamma_{W}\frac{D_{WW}}{H}{C_{W}}_{J}^{m+1}=0, \end{array}$$
where $C_{W}{}_{j}^{m+1}$ is the grid function of the concentration of water molecules.Subsystem 3 (26)–(28) was discretized on the time grid (32) and the extended non-uniform coordinate grid $G_{yH}$
(45)$$G_{yH}=\left\{y_{j}\left|\;j=1,\ldots ,J+2\right.\right\},$$
including the coordinate grid $G_{y}$ (33), supplemented by the first and last nodes, the coordinates of which correspond to the outer boundaries of metal electrodes
(46)$$s_{1}=0;$$
(47)$$\frac{s_{2}-s_{1}}{\Delta t}=0;$$
(48)$$\begin{array}{cc} \; & s_{m+1}=2\left(\frac{2-\Delta t^{2}\omega_{0m}^{2}}{\beta \Delta t+2}\right)s_{m}+\left(\frac{\beta \Delta t-2}{\beta \Delta t+2}\right)s_{m-1}+\frac{\omega_{0m}^{2}\Delta t^{2}L^{2}}{4\left(\beta \Delta t+2\right)}\times \\ \; & \times \left[\eta_{I}V_{I}\sum_{j=2}^{J}\;\left(\begin{array}{cc} \; & \left({C_{I}}_{j+1}^{m}-\frac{1}{2h}\underset{j=2}{\sum^{J}}\;\left({C_{I}}_{j+1}^{1}+{C_{I}}_{j}^{1}\right)\Delta y_{j}\right)\left(y_{j+1}-y_{\frac{J}{2}}\right)+\\ \; & +\left({C_{I}}_{j}^{m}-\frac{1}{2h}\underset{j=2}{\sum^{J}}\;\left({C_{I}}_{j+1}^{1}+{C_{I}}_{j}^{1}\right)\Delta y_{j}\right)\left(y_{j}-y_{\frac{J}{2}}\right) \end{array}\right)\Delta y_{j}\right.+\\ & \left.+\eta_{W}V_{W}\sum_{j=2}^{J}\;\left(\begin{array}{cc} \; & \left({C_{W}}_{j+1}^{m}-\frac{1}{2h}\underset{j=2}{\sum^{J}}\;\left({C_{W}}_{j+1}^{1}+{C_{W}}_{j}^{1}\right)\Delta y_{j}\right)\left(y_{j+1}-y_{\frac{J}{2}}\right)+\\ \; & +\left({C_{W}}_{j}^{m}-\frac{1}{2h}\underset{j=2}{\sum^{J}}\;\left({C_{W}}_{j+1}^{1}+{C_{W}}_{j}^{1}\right)\Delta y_{j}\right)\left(y_{j}-y_{\frac{J}{2}}\right) \end{array}\right)\Delta y_{j}\right]:\\ & :\frac{1}{2}\sum_{j=2}^{J+1}\;\left(E_{j+1}^{m}{\left(y_{j+1}-y_{\frac{J}{2}}\right)}^{2}+E_{j}^{m}{\left(y_{j}-y_{\frac{J}{2}}\right)}^{2}\right)\Delta y_{j}; \end{array}$$
(49)$$\omega_{0m}=\frac{\lambda }{L_{S}{}^{2}}\sqrt{\frac{w_{S}\sum_{j=1}^{J+1}\left(E_{j+1}^{m}{\left(y_{j+1}-y_{\frac{J}{2}}\right)}^{2}+E_{j}^{m}{\left(y_{j}-y_{\frac{J}{2}}\right)}^{2}\right)\cdot \Delta y_{j}}{2m_{Lm}}}\;;$$
(50)$$m_{Lm}=\frac{m_{D}}{L_{S}}+2\rho_{M}w_{S}H+\frac{M_{W}w_{S}}{2}\overset{J}{\underset{j=2}{\sum }}\left(C_{W}{}_{j+1}^{m}+C_{W}{}_{j}^{m}\right)\cdot \Delta y_{j},$$
where $s_{m}$ is the grid function of the tip displacement of the cantilever beam; $E_{j}^{m}$ is the grid function of the Young’s modulus; $\omega_{0m}$ is the grid function of the resonant frequency; $m_{Lm}$ is the grid function of the linear density of the beam.

The developed technique for the numerical solution of the system of equations (34)–(50) is presented as a block diagram in Figure 1. To solve subsystem 2 by the Newton-Raphson method, including the Nernst-Planck equations for ions and water molecules, the technique provides for the combination of the grid functions of the concentrations of ions $C_{I}{}_{j}^{m}$ and water molecules $C_{W}{}_{j}^{m}$ into a single column vector $C_{j}^{m}=\left[\begin{array}{c} C_{I}{}_{j}^{m}\\ C_{W}{}_{j}^{m} \end{array}\right],\;j=1,\ldots ,J.$ The grid function $U_{m}=f\left(t_{m}\right)$ specifies the voltage change at the beam electrodes with time.

Due to the nonlinearity of the system of equations (34)–(50), the problem is solved iteratively. The variable *k* in the block diagram in Figure 1 reflects the iteration index, $\sigma^{\left(k\right)}$ is the residual at *k*-th iteration determined by the values of the concentration vector at *k*-th and (*k* + 1)-th iterations $C_{j}^{m\left(k\right)}=\left[\begin{array}{c} C_{I}{}_{j}^{m\left(k\right)}\\ C_{W}{}_{j}^{m\left(k\right)} \end{array}\right];\;\;\;\;\;C_{j}^{m\left(k+1\right)}=\left[\begin{array}{c} C_{I}{}_{j}^{m\left(k+1\right)}\\ C_{W}{}_{j}^{m\left(k+1\right)} \end{array}\right],\;\;\;j=1,\ldots ,J.$

This technique is implemented as a specialized software created in the MATLAB programming environment.

## 4. Model Verification, Results, and Discussion

The proposed technique and software tools for the numerical implementation of model (34)–(50) allow for a detailed analysis of transients in the IPMC actuator polymer for an arbitrary mode of the control voltage oscillations on time, calculation the amplitude-frequency characteristics and dependences of the oscillation amplitude of the cantilever beam on the applied voltage amplitude at different degree of the polymer membrane humidification, taking into account the dependence of the dimensions and mass of the membrane on its humidification, and also taking into account the evaporation rates of ions and water molecules from the surface of metal electrodes.

In order to verify the considered model and the proposed technique for the numerical simulation of the IPMC actuator, a number of experimental studies have been performed.

### 4.1. Experimental Setup

The bench block diagram for the IPMC actuator investigation is presented in Figure 2. The investigated IPMC actuator was fixed with probes, through which the voltage was supplied from an Agilent 33500B Series waveform generator. The tip displacements of the IPMC actuator were recorded by an L-GAGE LG5A65PUQ laser displacement controller, from which information was transmitted to an Agilent DSO-X 3014A oscilloscope and then to a PC.

### 4.2. Numerical Simulation Results and Their Discussion—Comparison with Experimental Data

In order to verify the developed model (34)–(50), technique and software tools, the numerical simulation results of the IPMC actuator in a fairly wide range of amplitudes (peak-to-peak value *U_A_* = 0–5 V) and frequencies (*f* = 0.5–50 Hz) of the control voltage were obtained. The values of the physical constants and parameters of the IPMC actuator model used in the calculations are given in Table 1.

The numerical simulation results of transients in the polymer membrane of the IPMC actuator, obtained using the developed software tools for implementation the model (34)–(50), are presented in Figure 3, Figure 4, Figure 5, Figure 6, Figure 7, Figure 8, Figure 9, Figure 10, Figure 11, Figure 12, Figure 13, Figure 14, Figure 15, Figure 16, Figure 17, Figure 18, Figure 19 and Figure 20. These results were obtained on a coordinate grid containing 300 steps at a time grid step Δ*t* = 3 ms for a peak-to-peak control voltage *U_A_* = 5 V, harmonically changing in time with a frequency *f* = 1 Hz (Figure 3, Figure 4, Figure 5, Figure 6, Figure 7, Figure 8, Figure 9, Figure 10 and Figure 11) and *f* = 10 Hz (Figure 12, Figure 13, Figure 14, Figure 15, Figure 16, Figure 17, Figure 18, Figure 19 and Figure 20).

The numerical simulation results presented in Figure 3, Figure 4, Figure 5, Figure 6, Figure 7, Figure 8, Figure 9, Figure 10, Figure 11, Figure 12, Figure 13, Figure 14, Figure 15, Figure 16, Figure 17, Figure 18, Figure 19 and Figure 20 demonstrate the possibility of a detailed analysis of the dynamics of changes in time in the spatial distributions of the concentrations of ions and water molecules in the polymer membrane volume of the IPMC actuator.

The numerical simulation results of the time dependences of the control voltage *U*(*t*), the beam tip displacement *s*(*t*), and the acting force *F*(*t*) for control voltage frequencies of 0.5–40 Hz are presented in Figure 21, Figure 22, Figure 23, Figure 24 and Figure 25.

The analysis of the developed model (34)–(50) and the presented graphs show that the time behavior in the spatial distributions of the concentrations of ions and water molecules in the polymer membrane volume of the IPMC actuator depends in a complicated way on a sufficiently large number of parameters given in Table 1, but it is most determined by the ratio between the diffusion coefficients of ions and water molecules in the polymer and the control voltage frequency. It can be seen from Figure 3, Figure 4, Figure 5, Figure 6, Figure 7, Figure 8, Figure 9, Figure 10 and Figure 11 that at low control voltage frequencies *f* ≤ 1 Hz, wavelike changes in the concentrations have a significant amplitude throughout the entire volume of the polymer membrane, determining the noticeable on Figure 21 and Figure 22 deviations in the form of changes in the acting force *F*(*t*) and, accordingly, the form of the beam tip mechanical oscillations *s*(*t*) from the harmonic form.

According to Figure 12, Figure 13, Figure 14, Figure 15, Figure 16, Figure 17, Figure 18, Figure 19 and Figure 20, when the control voltage frequency increases, the amplitude of changes in the concentrations of ions and water molecules in the polymer membrane volume decreases markedly, which is mainly due to the inertia of diffusion processes. Moreover, a particularly strong decrease in the amplitude of concentration changes is observed in the central area of the membrane, where the spatial distributions of the concentrations during the entire transient remain almost uniform and even at a control voltage frequency *f* = 10 Hz change relative to equilibrium values by no more than 0.7% for water molecules and not more than 11% for cations (Figure 12, Figure 13, Figure 14, Figure 15, Figure 16, Figure 17, Figure 18, Figure 19 and Figure 20).

As a result, in accordance with Figure 23, Figure 24 and Figure 25, the change of the acting force in time *F*(*t*) with an increase in the control voltage frequency is more and more harmonic, and significant harmonic distortions of the form of the beam tip mechanical oscillations *s*(*t*) (see Figure 23b) reflect the interaction of the liquid and solid phases of the humidified actuator, taking into account the parameters that determine their inertia (diffusion coefficients *D_II_*, *D_WW_* for the liquid phase; elastic moduli *E_S_*, *E_M_*, densities and geometric dimensions of the beam layers for the solid phase; filtration coefficient *K*).

The result of the above-mentioned interaction of the liquid and solid phases of the humidified actuator is also a change in the phase shift between the control voltage oscillations and the beam tip mechanical oscillations, observed when increasing frequency in Figure 23, Figure 24 and Figure 25. If at a frequency *f* = 10 Hz and at lower frequencies the phase shift between the control voltage and the mechanical oscillations is approximately π/2 (Figure 21, Figure 22 and Figure 23), then at a frequency *f* = 36 Hz, which is close to the resonant frequency, the phase shift, in accordance with Figure 24, increases to π, and with a further increase in frequency to *f* = 40 Hz, in accordance with Figure 25, it reaches a value of 3π/5. It is important to note that the phase shift between the oscillations of the control voltage *U*(*t*) and the acting force *F*(*t*) remains unchanged and equal to approximately π/2 at any frequencies (Figure 21, Figure 22, Figure 23, Figure 24 and Figure 25).

To obtain experimental dependences of the displacement amplitude on the peak-to-peak control voltage, IPMC actuators with dimensions of 20 × 5 mm based on the Nafion 117 ion-exchange membrane with a thickness of 175 µm were used. The deposition of Pt electrodes on the membrane surface was carried out according to the technology described in [22,23]. The manufactured IPMC actuators were kept in deionized water for a day, after which they were investigated on the bench described in Section 4.1.

Figure 26, Figure 27 and Figure 28 show the calculated and experimental dependences of the beam tip displacement amplitude on the peak-to-peak control voltage with a frequency of 1 Hz (Figure 26), the beam tip displacement from the DC control voltage (Figure 27), as well as the calculated and experimental amplitude-frequency characteristics (AFC) of the IPMC actuator (Figure 28).

The analysis of the graphs presented in Figure 26, Figure 27 and Figure 28 indicates a good agreement between the calculated and experimental dependences of the beam tip displacement amplitude on the peak-to-peak control voltage (Figure 26) and the calculated and experimental AFC of the investigated IPMC actuator both in the beam mechanical oscillation amplitude and in the resonant frequency (*f_R_* ≈ 36 Hz), which corresponds to the characteristic maximum in Figure 28. In this case, the calculated dependence of the beam tip displacement on the DC control voltage level (Figure 27) gives a good agreement with the experiment only at control voltage *U_A_* ≤ 1.5 V, which is probably due to the influence of factors not considered in the model (34)–(50).

## 5. Conclusions

This article focuses on the development and numerical implementation of a mathematical model of an actuator based on ionic polymer-metal composite (IPMC). To simulate the most significant physical processes in the IPMC actuator such as transients of the spatial distributions of the ion and water molecule concentrations under the changing in time the electric field and the process of cantilever beam mechanical oscillations using comparatively small computing resources, multiphysics model, corresponding technique, and software tools for numerical simulation were developed. The developed IPMC actuator model is based on the thermodynamic theory of irreversible processes and includes the modified Nernst-Planck equations for both ions and water molecules, the Poisson equation, and the mechanical oscillation equation of a cantilever beam with a fixed end.

Using the developed numerical simulation software, the spatial distributions of the concentrations of ions and water molecules in the IPMC polymer membrane, transients of the force and the beam tip displacement at different values of amplitude and frequency of the control voltage, as well as amplitude-frequency characteristics of the actuator were calculated and investigated.

For verification of the developed numerical model and software, experimental characteristics of the IPMC actuator were measured. According to the comparative analysis, the numerical simulation results demonstrate good agreement with the experimental data.

The proposed numerical model of the IPMC actuator can be implemented in any programming language and allows for fast computational experiments.

## Figures and Tables

**Figure 1 micromachines-11-01119-f001:**
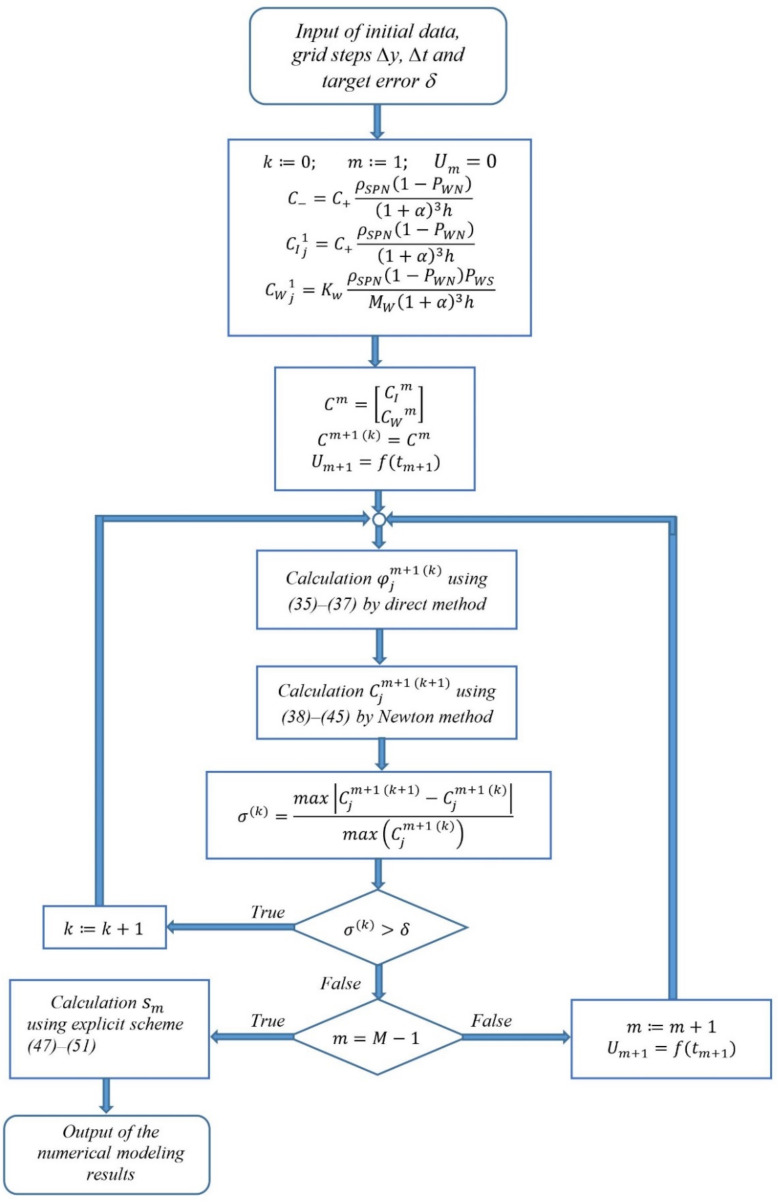
Technique for the numerical solution of system (35)–(51).

**Figure 2 micromachines-11-01119-f002:**
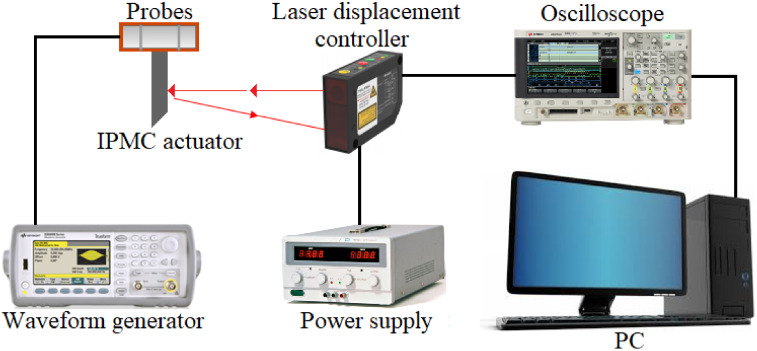
Bench block diagram for the IPMC actuator investigation.

**Figure 3 micromachines-11-01119-f003:**
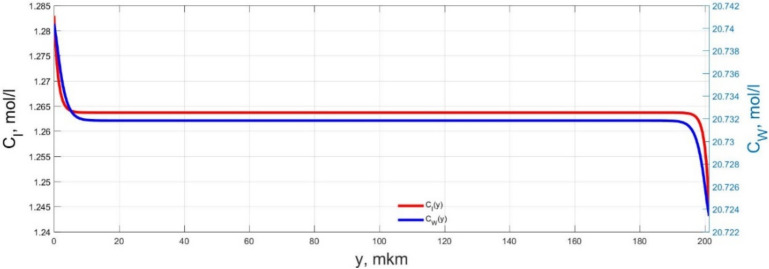
Spatial distributions of the concentrations of ions *C_I_*(*y*) and water molecules *C_W_*(*y*) in the IPMC polymer membrane at *U_A_* = 5 V and *f* = 1 Hz at a time point *t_m_* = 3 ms.

**Figure 4 micromachines-11-01119-f004:**
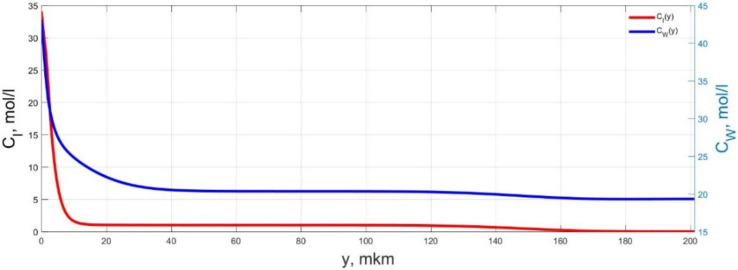
Spatial distributions of the concentrations of ions *C_I_*(*y*) and water molecules *C_W_*(*y*) in the IPMC polymer membrane at *U_A_* = 5 V and *f* = 1 Hz at a time point *t_m_* = 27 ms.

**Figure 5 micromachines-11-01119-f005:**
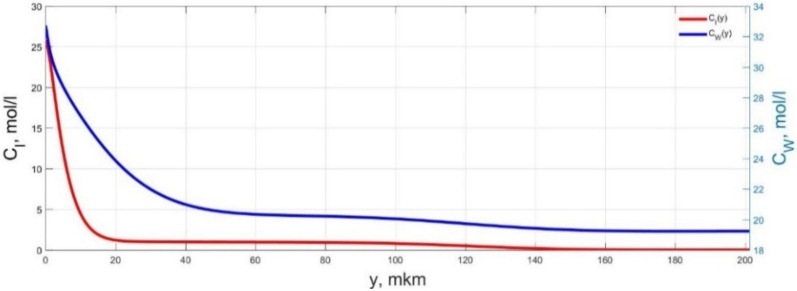
Spatial distributions of the concentrations of ions *C_I_*(*y*) and water molecules *C_W_*(*y*) in the IPMC polymer membrane at *U_A_* = 5 V and *f* = 1 Hz at a time point *t_m_* = 42 ms.

**Figure 6 micromachines-11-01119-f006:**
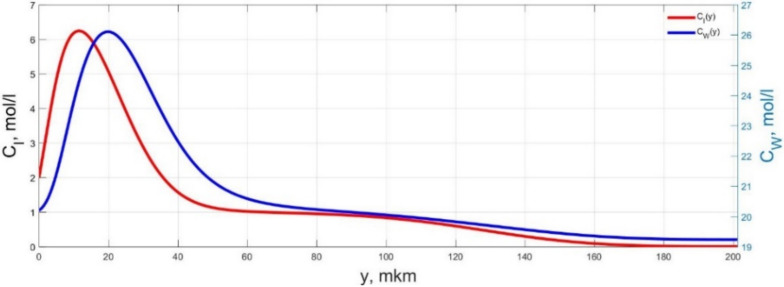
Spatial distributions of the concentrations of ions *C_I_*(*y*) and water molecules *C_W_*(*y*) in the IPMC polymer membrane at *U_A_* = 5 V and *f* = 1 Hz at a time point *t_m_* = 57 ms.

**Figure 7 micromachines-11-01119-f007:**
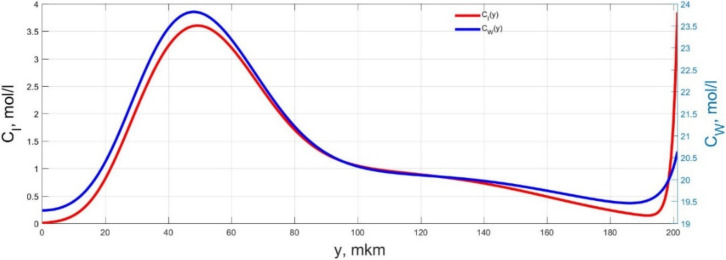
Spatial distributions of the concentrations of ions *C_I_*(*y*) and water molecules *C_W_*(*y*) in the IPMC polymer membrane at *U_A_* = 5 V and *f* = 1 Hz at a time point *t_m_* = 72 ms.

**Figure 8 micromachines-11-01119-f008:**
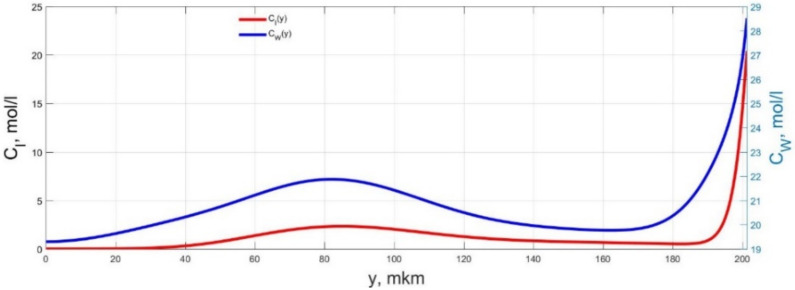
Spatial distributions of the concentrations of ions *C_I_*(*y*) and water molecules *C_W_*(*y*) in the IPMC polymer membrane at *U_A_* = 5 V and *f* = 1 Hz at a time point *t_m_* = 87 ms.

**Figure 9 micromachines-11-01119-f009:**
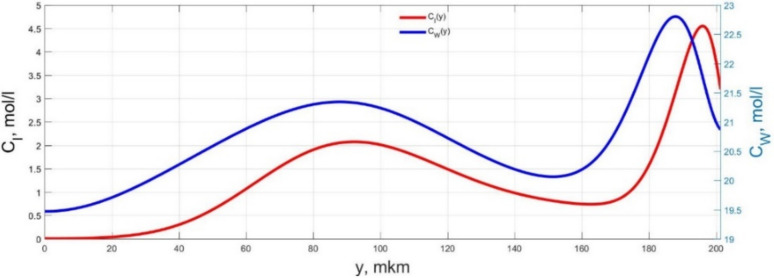
Spatial distributions of the concentrations of ions *C_I_*(*y*) and water molecules *C_W_*(*y*) in the IPMC polymer membrane at *U_A_* = 5 V and *f* = 1 Hz at a time point *t_m_* = 102 ms.

**Figure 10 micromachines-11-01119-f010:**
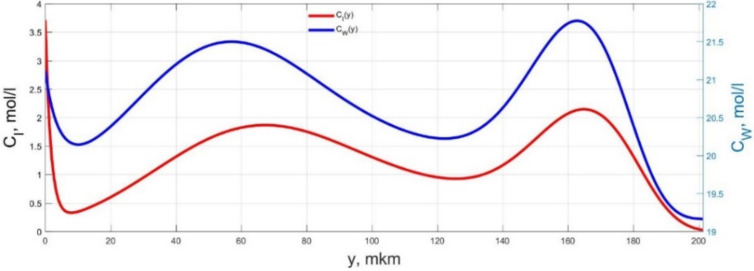
Spatial distributions of the concentrations of ions *C_I_*(*y*) and water molecules *C_W_*(*y*) in the IPMC polymer membrane at *U_A_* = 5 V and *f* = 1 Hz at a time point *t_m_* = 117 ms.

**Figure 11 micromachines-11-01119-f011:**
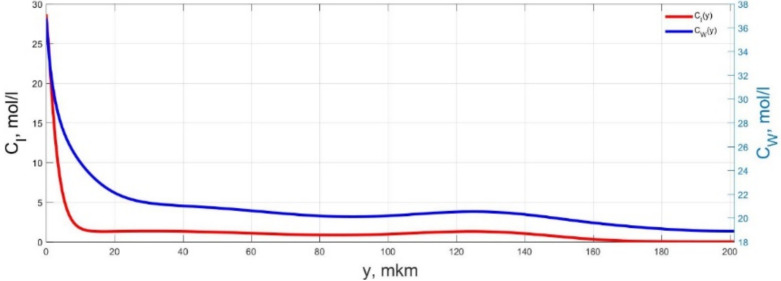
Spatial distributions of the concentrations of ions *C_I_*(*y*) and water molecules *C_W_*(*y*) in the IPMC polymer membrane at *U_A_* = 5 V and *f* = 1 Hz at a time point *t_m_* = 132 ms.

**Figure 12 micromachines-11-01119-f012:**
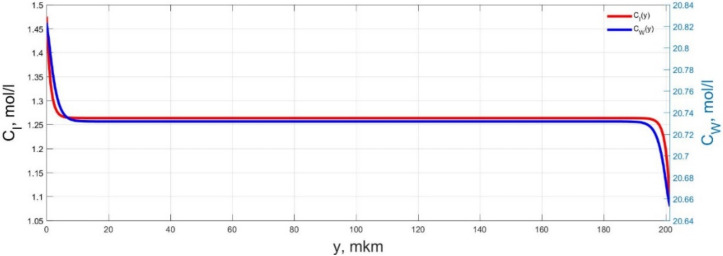
Spatial distributions of the concentrations of ions *C_I_*(*y*) and water molecules *C_W_*(*y*) in the IPMC polymer membrane at *U_A_* = 5 V and *f* = 10 Hz at a time point *t_m_* = 3 ms.

**Figure 13 micromachines-11-01119-f013:**
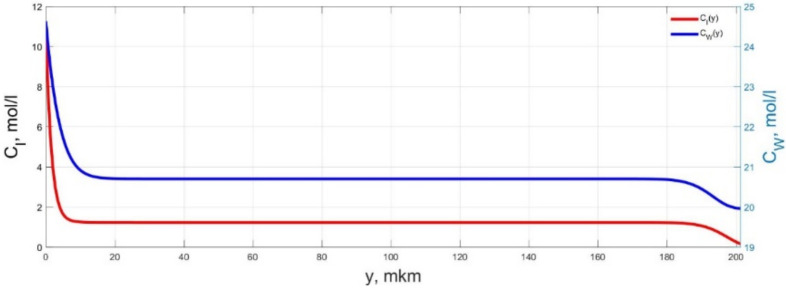
Spatial distributions of the concentrations of ions *C_I_*(*y*) and water molecules *C_W_*(*y*) in the IPMC polymer membrane at *U_A_* = 5 V and *f* = 10 Hz at a time point *t_m_* = 6 ms.

**Figure 14 micromachines-11-01119-f014:**
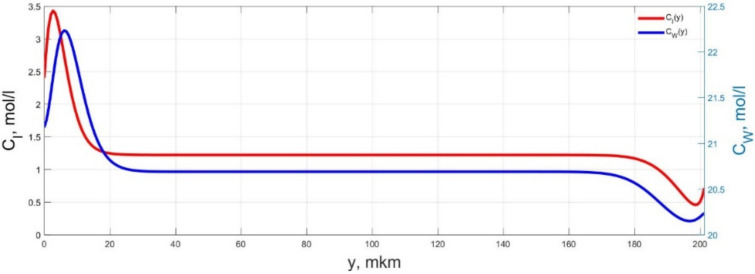
Spatial distributions of the concentrations of ions *C_I_*(*y*) and water molecules *C_W_*(*y*) in the IPMC polymer membrane at *U_A_* = 5 V and *f* = 10 Hz at a time point *t_m_* = 9 ms.

**Figure 15 micromachines-11-01119-f015:**
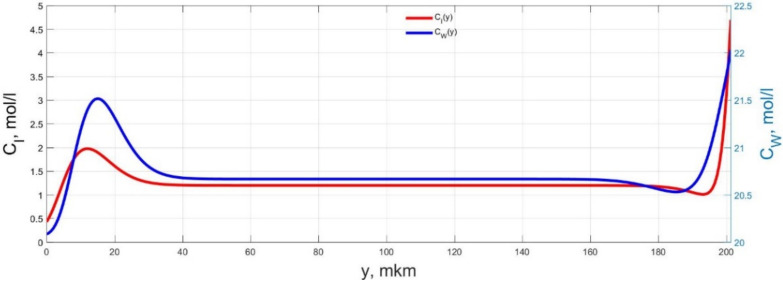
Spatial distributions of the concentrations of ions *C_I_*(*y*) and water molecules *C_W_*(*y*) in the IPMC polymer membrane at *U_A_* = 5 V and *f* = 10 Hz at a time point *t_m_* = 12 ms.

**Figure 16 micromachines-11-01119-f016:**
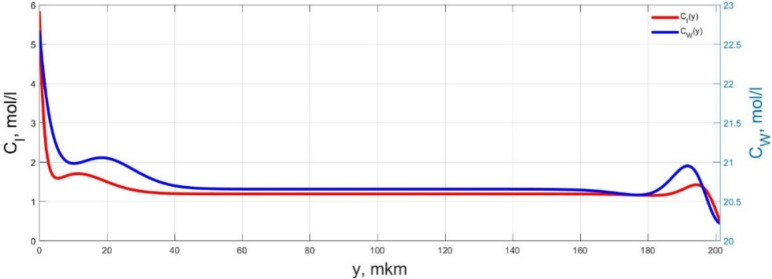
Spatial distributions of the concentrations of ions *C_I_*(*y*) and water molecules *C_W_*(*y*) in the IPMC polymer membrane at *U_A_* = 5 V and *f* = 10 Hz at a time point *t_m_* = 15 ms.

**Figure 17 micromachines-11-01119-f017:**
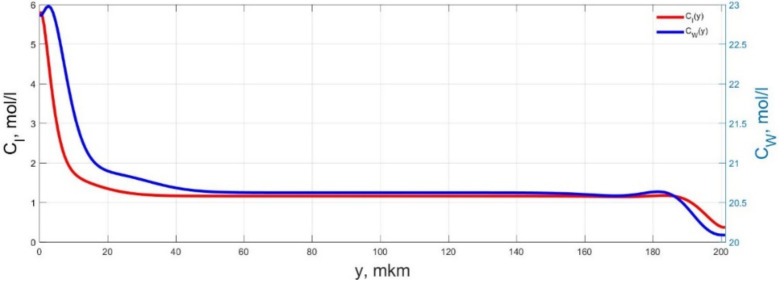
Spatial distributions of the concentrations of ions *C_I_*(*y*) and water molecules *C_W_*(*y*) in the IPMC polymer membrane at *U_A_* = 5 V and *f* = 10 Hz at a time point *t_m_* = 18 ms.

**Figure 18 micromachines-11-01119-f018:**
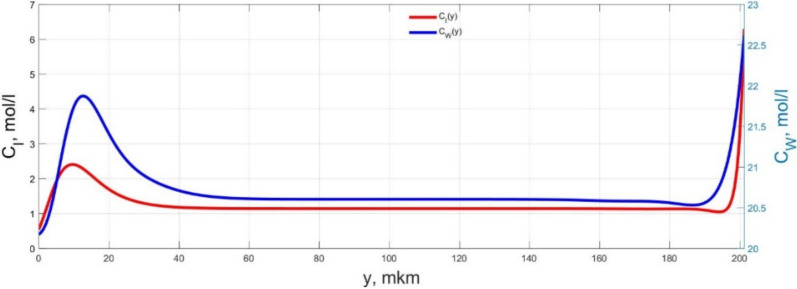
Spatial distributions of the concentrations of ions *C_I_*(*y*) and water molecules *C_W_*(*y*) in the IPMC polymer membrane at *U_A_* = 5 V and *f* = 10 Hz at a time point *t_m_* = 21 ms.

**Figure 19 micromachines-11-01119-f019:**
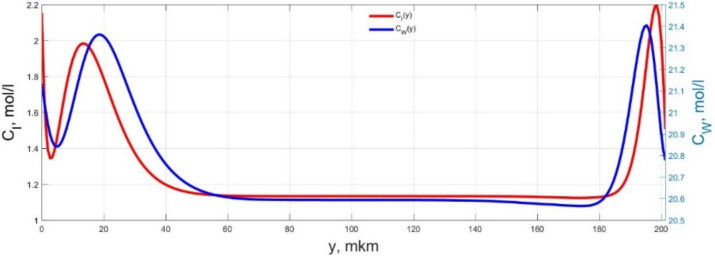
Spatial distributions of the concentrations of ions *C_I_*(*y*) and water molecules *C_W_*(*y*) in the IPMC polymer membrane at *U_A_* = 5 V and *f* = 10 Hz at a time point *t_m_* = 24 ms.

**Figure 20 micromachines-11-01119-f020:**
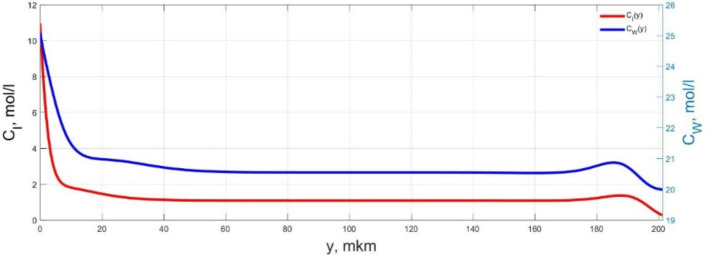
Spatial distributions of the concentrations of ions *C_I_*(*y*) and water molecules *C_W_*(*y*) in the IPMC polymer membrane at *U_A_* = 5 V and *f* = 10 Hz at a time point *t_m_* = 27 ms.

**Figure 21 micromachines-11-01119-f021:**
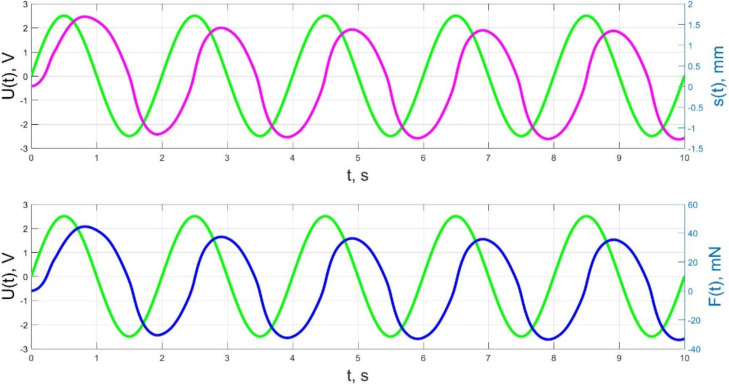
Transients in the IPMC actuator at *U_A_* = 5 V and *f* = 0.5 Hz: green lines are the control voltage *U*(*t*); pink lines are the beam tip displacement *s*(*t*); blue lines are the force *F*(*t*).

**Figure 22 micromachines-11-01119-f022:**
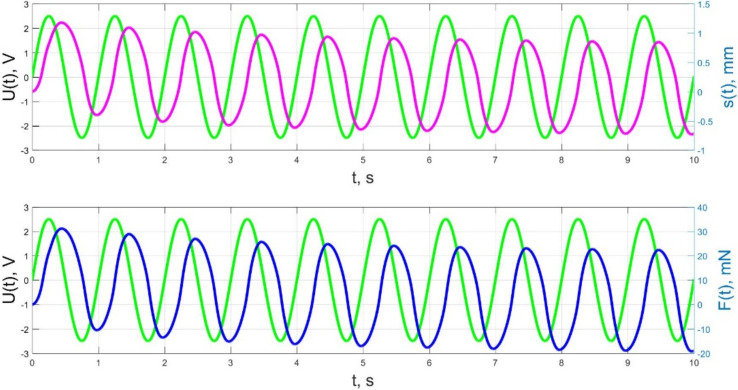
Transients in the IPMC actuator at *U_A_* = 5 V and *f* = 1 Hz: green lines are the control voltage *U*(*t*); pink lines are the beam tip displacement *s*(*t*); blue lines are the force *F*(*t*).

**Figure 23 micromachines-11-01119-f023:**
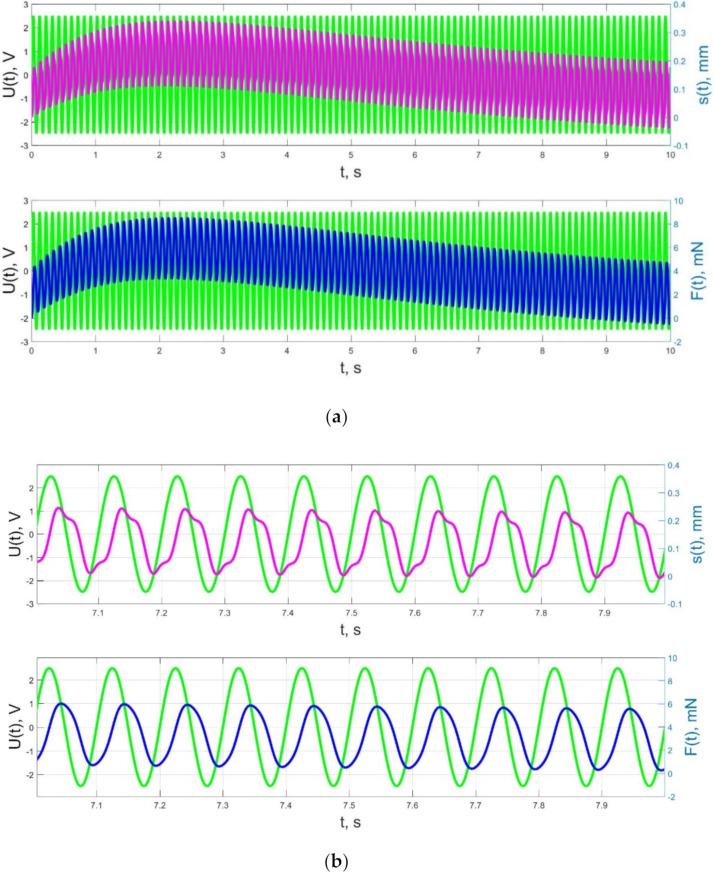
Transients in the IPMC actuator at *U_A_* = 5 V and *f* = 10 Hz at intervals t = 0–10 s (**a**) and t = 7–8 s (**b**): green lines are the control voltage *U*(*t*); pink lines are the beam tip displacement *s*(*t*); blue lines are the force *F*(*t*).

**Figure 24 micromachines-11-01119-f024:**
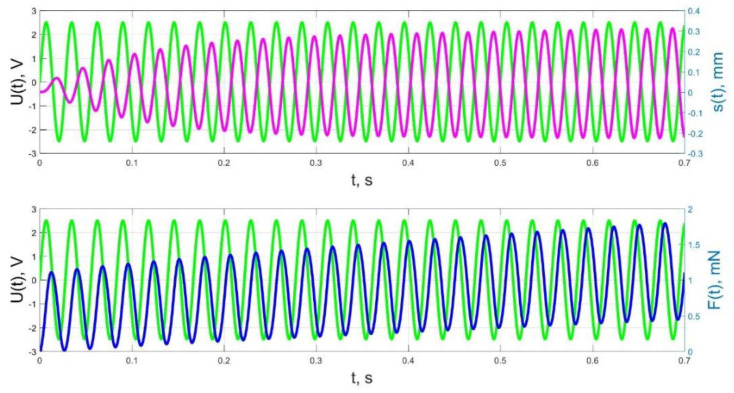
Transients in the IPMC actuator at *U_A_* = 5 V and *f* = 36 Hz: green lines are the control voltage *U*(*t*); pink lines are the beam tip displacement *s*(*t*); blue lines are the force *F*(*t*).

**Figure 25 micromachines-11-01119-f025:**
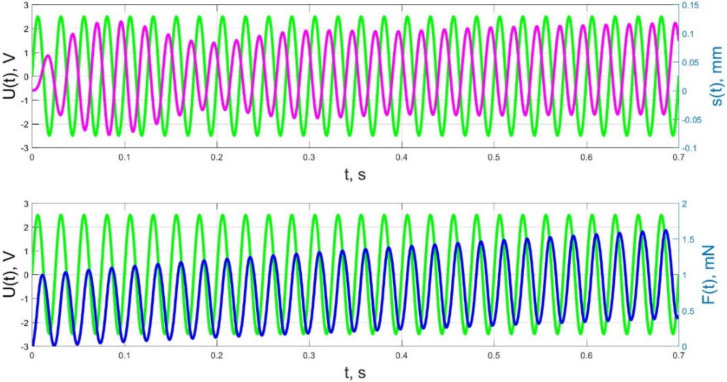
Transients in the IPMC actuator at *U_A_* = 5 V and *f* = 40 Hz: green lines are the control voltage *U*(*t*); pink lines are the beam tip displacement *s*(*t*); blue lines are the force *F*(*t*).

**Figure 26 micromachines-11-01119-f026:**
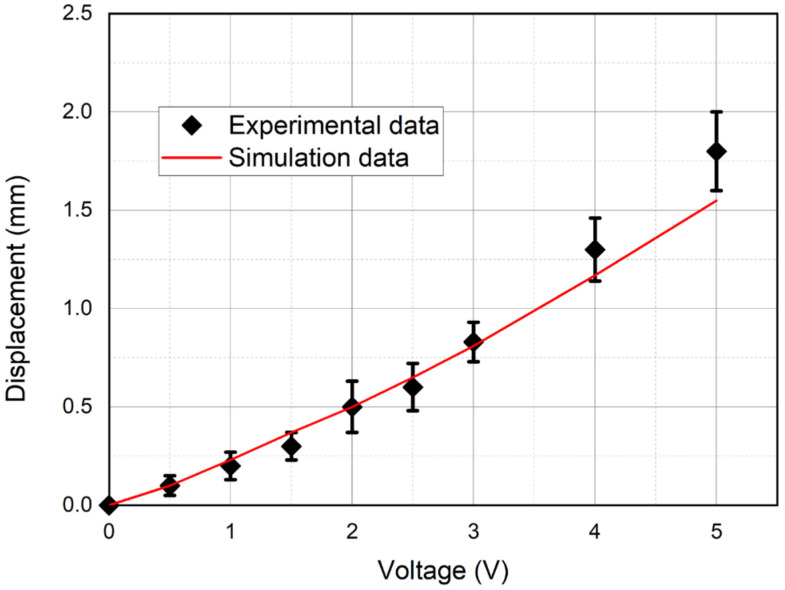
The calculated and experimental dependences of the beam tip displacement amplitude on the peak-to-peak control voltage at a frequency *f* = 1 Hz.

**Figure 27 micromachines-11-01119-f027:**
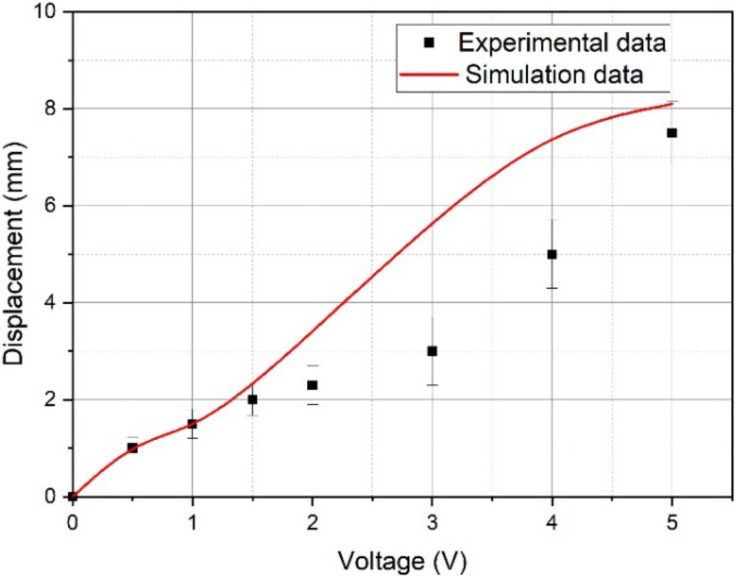
The calculated and experimental dependences of the beam tip displacement amplitude on the DC control voltage.

**Figure 28 micromachines-11-01119-f028:**
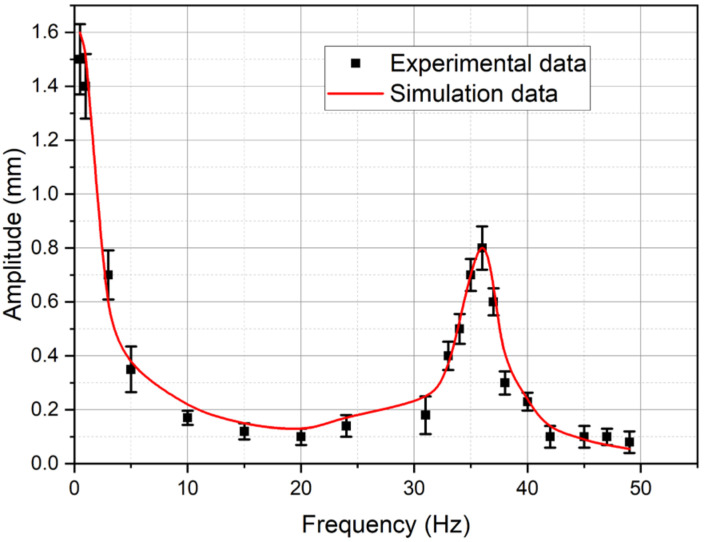
The calculated and experimental amplitude-frequency characteristics (AFC) of the IPMC actuator.

**Table 1 micromachines-11-01119-t001:** Parameters of the IPMC actuator model.

Parameter	Symbol	Value	Unit
Polymer trademark	–	Nafion N117	–
Length of the dry beam ^1^	*L*	15	mm
Width of the dry beam ^1^	*w*	5	mm
Thickness of the dry beam ^1^	*H*	183	μm
Thickness of metal electrodes	*H*	5	μm
Temperature	*T*	293	K
Diffusion coefficient of cations	*D_II_*	5.3 × 10^−6^	cm^2^⋅s^−^^1^
Diffusion coefficient of water molecules	*D_WW_*	3.87 × 10^−6^	cm^2^⋅s^−^^1^
Concentration of ions in the polymer	*C _+_*	0.9	mol⋅kg^−^^1^
Molar volume of ions	*V_I_*	−5.4	cm^3^⋅mol^−^^1^
Molar volume of water	*V_W_*	18	cm^3^⋅mol^−^^1^
Filtration coefficient	*K*	3.4 × 10^−14^	cm^2^⋅Pa^−^^1^⋅s^−^^1^
Elementary charge	*q*	1.6 × 10^−19^	C
Faraday constant	*F*	96,485	C⋅mol^−^^1^
Gas constant	*R*	8.31	J⋅K^−1^⋅mol^−1^
Permittivity of vacuum	*ε* _0_	8.85 × 10^−14^	F⋅cm^−^^1^
Relative permittivity of water	*ε*	81	–
Expansion coefficient of the membrane at maximum humidification	*α*	0.1	–
Relative charge of ion	*Z_I_*	1	–
Number of water molecules associated with one cation	*n_dW_*	1	–
Mass fraction of water in the dry polymer ^1^	*P_WN_*	0.05	–
Mass fraction of water in the humidified polymer	*P_WS_*	0.38	–
Young’s modulus of the dry polymer ^1^	*E_N_*	249	MPa
Young’s modulus of the humidified polymer	*E_S_*	114	MPa
Young’s modulus of metal electrodes	*E_M_*	23	GPa
Empirical coefficient	*η_I_*	200	MPa
Empirical coefficient	*η_W_*	200	MPa
Coefficient depending on the mode of beam bending oscillations	*λ*	3.52	–
Coefficient characterizing dissipative processes	*β*	19	s^−^^1^
Empirical coefficient determining the evaporation rate of cations into the external environment	*γ_I_*	0	–
Empirical coefficient determining the evaporation rate of water molecules into the external environment	*γ_W_*	0	–
Layer density of the dry membrane ^1^	*ρ_SPN_*	3.6 × 10^−2^	g·cm^−2^
Density of the electrode material	*ρ_M_*	21.5	g·cm^−3^
Molar mass of water	*M_W_*	18.01528	g⋅mol^−^^1^

^1^ At normal air humidity.

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
