# Peer review of "Multiphysics Simulator for the IPMC Actuator: Mathematical Model, Finite Difference Scheme, Fast Numerical Algorithm, and Verification"

_micromachines, 2020, doi:10.3390/mi11121119_

Round 1

Reviewer 1 Report

The research reported here may have some impact on the future understanding of IPMC. There are a few critical issues that have to be addressed before publish the manuscript.

1) Introduction: there was a new model introduced in Journal of Advanced Dielectrics, 7(2) 1720002 (2017). The new model was validated using the experimental results from some materials (Actuators, 7(4), 72 (2018)). This new model was completely missed from the manuscript. 

2) In the research of IPMC, most of the results were reported in terms of displacement versus time for an IPMC under a DC voltage. To validate the theoerical results by others, the results of displacement versus time for an IPMC under a constant DC voltage should be concluded from the theoretical study here and the results should be presented here in the manuscript. If possible, it would be better to have different DC voltage to determine whether the strength of the DC has some influences on the time dependence of the displacement.

Author Response

The authors are very grateful to the Reviewer for important and useful comments on the manuscript.

Point 1: Introduction: there was a new model introduced in Journal of Advanced Dielectrics, 7(2) 1720002 (2017). The new model was validated using the experimental results from some materials (Actuators, 7(4), 72 (2018)). This new model was completely missed from the manuscript. 

Response 1: References on a new model introduced in Journal of Advanced Dielectrics, 7(2) 1720002 (2017) and validated using the experimental results from Actuators, 7(4), 72 (2018) are included in lines 71-72 of the manuscript.

Point 2: In the research of IPMC, most of the results were reported in terms of displacement versus time for an IPMC under a DC voltage. To validate the theoretical results by others, the results of displacement versus time for an IPMC under a constant DC voltage should be concluded from the theoretical study here and the results should be presented here in the manuscript. If possible, it would be better to have different DC voltage to determine whether the strength of the DC has some influences on the time dependence of the displacement.

Response 2: The proposed multiphysics simulator model and finite difference scheme are valid for both DC and AC terminal voltages. To validate the theoretical results Figures 26–28 show the calculated and experimental dependences of the beam tip displacement for AC voltage (Figures 26 and 28) as well as for DC voltage (Figure 27). Figure 26 shows a dependence of the displacement amplitude on the AC control voltage with a frequency of 1 Hz. Figure 27 illustrates a dependence of the displacement on the DC control voltage (for the DC voltage band of 0 to 5 V). Figure 28 reflects the calculated and experimental amplitude-frequency characteristics of the IPMC actuator (for the frequency band of 0 to 50 Hz).

Reviewer 2 Report

The authors formulated a finite difference scheme for calculating the motion of IPMC actuators.  They also demonstrated the calculation by comparing with a set of experimental data.  It is nice to know that finite difference scheme is still useful while nowadays the finite element method seems to be the standard scheme for studying the differential equations. 

The authors used a linear approximation in many places in the formulation.  I would say most of the linear approximations are quite reasonable, but I would be rather careful in making the approximation of Eq. 24, where the pressure is set proportional to the density of the liquid phase.  Although such an approximation may be allowed at the level of the present study, it would be helpful for the readers if some explanation about the physics behind that approximation is given in the text. 

Author Response

The authors are very grateful to the Reviewer for important and useful comments on the manuscript.

Point 1: The authors used a linear approximation in many places in the formulation.  I would say most of the linear approximations are quite reasonable, but I would be rather careful in making the approximation of Eq. 24, where the pressure is set proportional to the density of the liquid phase.  Although such an approximation may be allowed at the level of the present study, it would be helpful for the readers if some explanation about the physics behind that approximation is given in the text. 

Response 1: The authors agree with the Reviewer. This important question was considered in some publications such as

Takagi, K.; Shahinpoor, M.; Schneider, H.-J.; Oh, I.-K.; Porfiri, M.; Kim, K.J.; Johanson, U.; Vunder, V.; Branco P.J.C.; Tan, X.; Leang, K.; Lumia, R. Ionic Polymer Metal Composites (IPMCs): Smart Multi-Functional Materials and Artificial Muscles, Volume 1, 1st ed.; Shahinpoor, M., Ed.; Royal Society of Chemistry: Cambridge, UK, 2015; ISBN 978-1-78262-258-1.

Caponetto, R.; De Luca, V.; Di Pasquale, G.; Graziani, S.; Sapuppo, F.; Umana, E. A new multi-physics model of an IP2C actuator in the electrical, chemical, mechanical and thermal domains. In Proceedings of the 2013 IEEE International Instrumentation and Measurement Technology Conference (I2MTC), 2013; pp. 971-975.

References on the linear approximation like (24) are included in line 193 of the manuscript.

Round 2

Reviewer 1 Report

It looks good to this reviewer. The manuscript may be accepted for this current form. BUT it would be much better if the authors can simplify the results
(formulas) for others to easily adopt it.